# Nursing Students’ Knowledge Among Healthcare-Associated Infections: A Systematic Review

**DOI:** 10.3390/ijerph22111609

**Published:** 2025-10-22

**Authors:** Vincenza Giordano, Chiara Palazzo, Caterina Mercuri, Vittoria Verde, Teresa Rea, Patrizia Doldo, Assunta Guillari

**Affiliations:** 1Department of Public Health, University of Naples “Federico II”, 80131 Naples, Italy; vincenza.giordano@unina.it (V.G.); teresa.rea@unina.it (T.R.); 2Pediatric Oncology, AORN Santobono-Pausilipon, 80122 Naples, Italy; chiara.palazzo@students.uniroma2.eu; 3Clinical and Experimental Medicine Department, Magna Graecia University, 88100 Catanzaro, Italy; doldo@unicz.it; 4University of Naples “Federico II”, 80131 Naples, Italy; vitto.verde@studenti.unina.it; 5Department of Translational Medical Sciences, Clinical Research Center DEMeTra, University of Naples “Federico II”, 80131 Naples, Italy; assunta.guillari@unina.it

**Keywords:** healthcare-associated infections, nursing students, knowledge, competence, perception, infection prevention and control, standard precautions, hand hygiene

## Abstract

Background: Healthcare-associated infections represent a significant public health problem, with a major impact on patient safety and quality of care. Nursing students play a central role in implementing infection prevention and control measures, yet the existing literature highlights gaps in their preparedness. Objective: To investigate nursing students’ knowledge of healthcare-associated infections, providing a comprehensive understanding of their level of awareness and identifying potential gaps that could guide improvements in nursing education. Methods: A systematic review was conducted by PRISMA guidelines. Searches were performed in PubMed, CINAHL Complete, APA PsycArticles, and APA PsycInfo, using predefined keywords and inclusion criteria. Twenty-two studies met the eligibility requirements. The articles were assessed for methodological quality using validated appraisal tools. Results: Across the included studies, nursing students demonstrated good knowledge of certain infection prevention measures, particularly standard precautions and hand hygiene. However, significant theoretical gaps were identified, especially regarding epidemiology, transmission mechanisms, and risk factors for healthcare-associated infections. Knowledge tended to improve with academic progression, clinical experience, and the use of active, contextualized teaching strategies. A consistent gap between theoretical understanding and practical application was observed. Conclusions: Nursing curricula should systematically integrate theory and practice, ensure alignment with validated international guidelines, and adopt active, practice-oriented teaching approaches. Strengthening these areas could improve nursing students’ infection prevention competencies, thereby enhancing patient safety and quality of care.

## 1. Introduction

Healthcare-associated infections (HAIs) represent a persistent and growing challenge for global public health, significantly affecting the quality of care and patient safety. Defined as infections acquired during the process of care in a hospital or healthcare setting, which were not present or incubating at the time of admission, HAIs are among the most frequent adverse events in healthcare [1,2].

The World Health Organization (WHO) has identified HAIs as a critical issue, highlighting their role in causing long-term disabilities, increasing hospitalization rates, and contributing to the phenomenon of antimicrobial resistance [3,4,5,6]. These infections result in significant social and economic burdens, emphasizing the urgent need for effective prevention and control measures (IPC) [7].

Globally, the prevalence of HAIs varies significantly based on geographic region, healthcare infrastructure, and resource availability. The WHO estimates that approximately 15% of hospitalized patients worldwide are affected by HAIs, with the highest prevalence reported in low- and middle-income countries [8]. Recent studies have confirmed that the global prevalence of HAIs is increasing, reaching up to 27% in some regions of Africa, while in Europe the average prevalence is 6.5% in acute care settings [9]. In Italy, the prevalence of HAIs among hospitalized patients has risen steadily, increasing from 4.9% in 2016 to 6.3% in 2022, exceeding the European average of 5.7% [10]. This upward trend highlights the urgent need for improved infection control practices and targeted interventions.

HAIs encompass a diverse range of conditions, including surgical site infections, ventilator-associated pneumonia, urinary tract infections, and catheter-associated sepsis [11]. These conditions carry significant implications for morbidity, mortality, and healthcare costs [12,13]. Therefore, the prevention of HAIs is a cornerstone of quality healthcare, with nurses playing a central role due to their direct and continuous interaction with patients [14]. Nurses are pivotal in implementing infection prevention strategies, such as hand hygiene, proper use of personal protective equipment (PPE), and early detection of infections at device sites [14,15]. Their adherence to infection control measures is critical to breaking the chain of transmission and ensuring patient safety. However, studies reveal significant gaps in adherence to infection control practices among nurses, including inconsistent hand hygiene and improper glove use, which increase the risk of infection spread [15,16,17]. Access to clear guidelines, appropriate training, and adequate resources has been shown to improve compliance, to increase healthcare workers’ adherence to infection control measure and reduce infection rates [18,19].

The WHO emphasizes the importance of integrating IPC education into undergraduate nursing curricula, aiming to equip students with the necessary knowledge and skills to implement evidence-based practices in clinical settings [20]. Parreira et al. [21], reinforce this view, highlighting that early and structured IPC education can enhance students’ preparedness for clinical practice. Despite this, existing literature reveals substantial gaps in the knowledge of HAIs among nursing students, often reflecting insufficient preparation during their training, as reported Labrague et al. [22], that identified disparities in knowledge and adherence to IPC measures, particularly about hand hygiene and appropriate use of PPE.

One of the main challenges in addressing these knowledge gaps is the lack of uniformity in nursing curricula across Europe. In fact, Salinen et al. [23] and Taneva et al. [24] have highlighted significant variability in the structure and content of nursing education programs, with no standardized recommendations for courses dedicated to the prevention and control of HAIs. This inconsistency hampers the ability of nursing students to acquire a comprehensive understanding of IPC measures, limiting their preparedness for clinical practice.

Some reviews in the literature, such as those by Ojo and Ojo [25] and Da Silva et al. [26], have primarily focused on individual aspects of IPC, such as hand hygiene, while neglecting the broader knowledge gaps within nursing education.

These limitations highlight the need for a more integrated approach to evaluate and improve nursing students’ overall understanding of HAIs.

This systematic review distinguishes itself by adopting a comprehensive and updated approach to analyzing the knowledge of nursing students regarding HAIs. Unlike previous reviews, which focused on fragmented aspects of IPC, this work considers the overall understanding of HAIs among nursing students, including specific practices like hand hygiene and standard precautions as part of a broader educational context. Furthermore, this review incorporates recent studies not available in earlier analyses, providing a more accurate and contemporary understanding of the topic.

### Objective of the Study

The purpose of this systematic review is to investigate nursing students’ knowledge of HAIs, aiming to provide a comprehensive understanding of their level of awareness and identify potential gaps in knowledge that could inform improvements in nursing education.

## 2. Materials and Methods

### 2.1. Study Design

This systematic review was conducted between February 2025 and October 2025 according to the Preferred Reporting Items for Systematic Reviews and Meta-Analysis (PRISMA) guidelines [27].

### 2.2. Search Methods

This systematic review addresses the following research question, formulated using the Population, Exposure, and Outcome (PEO) framework: “What knowledge do nursing students have regarding healthcare-associated infections?”

The PEO framework defines the population under study, the exposure or phenomenon of interest, and the outcomes assessed (Table 1).

Two members of the team independently conducted electronic searches using the following databases: PubMed, CINAHL Complete, APA PsycArticles, and APA PsycInfo. The main search terms used were “nursing students,” “student nurses,” “undergraduate nursing students,” “pre-licensure nurse,” “cross infection,” “hospital-acquired infections,” “healthcare-associated infections,” “infection prevention,” “knowledge,” “education,” “understanding,” “awareness”, “competence”, “attitude” and “perception” Search strings were constructed using Boolean operators “OR” and “AND”, combining the terms to optimize the sensitivity and specificity of the literature search.

No time limits or other filters have been applied to the search.

The resulting search strings tailored to each database are detailed below (Table 2, Table 3 and Table 4):

### 2.3. Inclusion and Exclusion Criteria

The following inclusion and exclusion criteria were established for the literature search.

Inclusion criteria:(a)Primary studies;(b)Original studies;(c)Literature reviews;(d)Articles that evaluated nursing students’ understanding of healthcare-associated infections;(e)Articles in English.

Exclusion criteria:(a)Guidelines;(b)Dissertations, oral presentations, methodological or theoretical descriptions or individual clinical cases;(c)Studies that did not evaluate nursing students’ understanding of healthcare-associated infections;(d)Articles written in languages other than English.

### 2.4. Search Outcome

The selection process initially identified 1513 articles, of which 1303 came from PubMed and 190 from the CINAHL Complete and 20 from APA PsycArticles and APA PsycInfo databases. All records were uploaded and managed through Rayyan (Rayyan Systems Inc., Cambridge, MA, USA), an intelligent systematic review software designed to facilitate the selection process and minimize the risk of bias. Initial screening by title was performed, eliminating duplicates (*n* = 95). This phase was conducted independently by two reviewers (VG and CM); any conflicts were resolved through consensus discussion. After removing duplicates, the titles and abstracts of 1418 articles were read, and 1362 articles were eliminated for various reasons. In the end, 56 articles were selected for full-text reading. During this phase, 34 articles were excluded for several reasons (wrong outcome, unable to extract the sample of nurses, wrong population, wrong type publication). Consequently, 22 studies were included in the final review. The entire process of identification, screening, eligibility and inclusion was documented and represented graphically according to the PRISMA 2020 flowchart (Figure 1) [27].

### 2.5. Quality Appraisal

The methodological quality of the included studies was assessed using the Joanna Briggs Institute (JBI) Critical Appraisal Checklists for cross-sectional and descriptive studies [28] and the National Institutes of Health [29] quality assessment tool for observational cohort and cross-sectional studies. The results of the appraisal are presented in Appendix A.

Among the cross-sectional studies, four achieved a high-quality rating, meeting ≥ 80% of the JBI criteria [30,31,32,33]. These studies demonstrated clear inclusion criteria, comprehensive description of subjects and settings, valid and reliable measurement of exposures and outcomes, and appropriate statistical analyses.

Several studies were rated as of moderate quality, with scores ranging between 6/8 and 7/8 on the JBI checklist. These included Alriyami et al. [34], Brosio et al. [35], Laiba Mazhar et al. [36], Syed and Al-Rawy [37], Shrestha et al. [38], Sharma and Bachani [39], Thakker and Jadhav [40], Al-Rawajfah and Tubaishat [41], Ojulong et al. [42], Blomgren et al. [43], Khubrani et al. [44], Bello et al. [45], Mitchell et al. [46], and Gould and Drey [47]. The main limitations were insufficient reporting of strategies to control for confounding variables, absence of sample size justification, and limited details on participant selection.

The single descriptive study by Colosi [48] was rated as moderate quality, with a score of 6/8. The study clearly described inclusion criteria and outcome measures but provided limited detail on the sampling frame and lacked strategies to address potential confounders.

Three studies reached the maximum JBI score of 8/8 and were thus rated as high quality: Darawad and Al-Hussami [49], Wu et al. [50], and Rahiman et al. [51]. These consistently applied standard criteria for measurement, identified and controlled for confounders, and performed robust statistical analyses.

When evaluated with the NIH quality assessment tool, scores ranged from 6/14 to 10/14, reflecting considerable variability in methodological rigor. The strongest results were observed in Rahiman et al. [51] and Darawad and Al-Hussami [49], both rated as good quality (10/14). These studies provided a clear research question, reliable measurement of exposures and outcomes, high participation rates, and some adjustment for confounding factors. Conversely, studies such as Gould and Drey [47], Blomgren et al. [43], and Al-Rawajfah and Tubaishat [41] were rated lower (6–7/14) due to the absence of sample size justification, lack of blinding, and limited adjustment for confounders.

All the evaluated articles were included in the present review. Despite some differences in the scores obtained using the NIH Quality Assessment Tool for Observational Cohort and Cross-Sectional Studies and the methodological quality checklist (JB), none of the studies showed a risk of bias significant enough to compromise their overall validity or relevance to the research question.

All studies clearly defined inclusion criteria, provided an adequate description of participants and settings, and measured exposure and outcomes in a valid and reliable manner. Even in cases where confounding factors or statistical control strategies were not explicitly reported, the overall methodological quality can be considered moderate to good, with scores ranging from 6/8 (75%) to 8/8 (100%) in the first assessment, and from 6/14 to 10/14 in the second.

The decision to include all articles is justified by the fact that each study contributes significantly to understanding the topic under review, providing coherent and complementary data aligned with the research objectives. Furthermore, the diversity of geographical, temporal, and methodological contexts represents an added value, as it allows for a broader and more comparative understanding of the phenomenon under investigation.

### 2.6. Data Synthesis

Although the included studies presented quantitative data, a meta-analysis was not appropriate due to considerable heterogeneity in study design, outcome measures, and knowledge scoring systems. Different instruments (e.g., ICSQ, WHO-based, or custom questionnaires) were used, with results expressed variably (e.g., scores out of 10, 25, or 30; percentages: mean ranks), and without a consistent definition of “adequate knowledge”. Furthermore, the included studies assessed different domains (e.g., knowledge only vs. knowledge and practices), further limiting the comparability of outcomes. These methodological discrepancies precluded meaningful statistical pooling. Therefore, a narrative synthesis was conducted, in line with PRISMA 2020 recommendations [27].

## 3. Results

### Characteristics of the Studies

This systematic review included eight articles, selected based on originality, relevance, and methodological rigor. The studies comprised nineteen cross-sectional studies [30,31,32,34,35,36,37,38,39,40,41,42,43,44,45,46,49,50,51] two descriptive study [43,48] and one quantitative observational pre-post study.

The research was conducted in a variety of geographical settings. Studies were carried out in developed countries such as Italy [35,48], France [32,33], Sweden [43], the United Kingdom [47], Australia [46], and Taiwan [50]. Research in developing countries included studies from Iran [31], Pakistan [36], Saudi Arabia [37,44], Jordan [41,49], Ghana [45], Namibia [42], Nepal [38], South Africa [51], and India [39] and Asian country.

The Infection Control Standardized Questionnaire (ICSQ), originally developed by Tavolacci et al. [32], was later used in various studies [30,32,34,35,45,48]

Others adopted structured questionnaires based on WHO and CDC guidelines [31,36]. Some studies relied on the WHO Hand Hygiene Knowledge Questionnaire [40,43], the Infection Control Evaluation (ICE) tool [50], or ad hoc instruments developed for the specific study aims [37,38,39,42,46,47,49,51]. The French multicenter study by Bouget Mohammedi et al. [33] combined the WHO hand hygiene tool with the Tavolacci standard precautions questionnaire and a survey of teaching strategies [32].

The characteristics of the studies were aggregated and summarized in summary Tables (Appendix A) to provide an organized and clear overview of the main findings. All studies highlighted the crucial role of nursing students’ education in preventing and controlling HAIs in clinical settings. Consequently, there is a growing need to strengthen training and competencies during undergraduate nursing education.

The cross-sectional study by Tavolacci et al. [32] investigated the knowledge levels of nursing, medical, physiotherapy, and radiology students regarding infection control and the most effective sources of information on the topic. Three areas of knowledge were examined: HAIs, hand hygiene (HH), and standard precautions (SP). Nursing students achieved the highest total score—23.2 (SD ± 2.35)—compared with physiotherapy students (21.9, SD ± 2.36), medical students (21.1, SD ± 2.35), and radiology students (20.5, SD ± 3.04) (*p* < 0.001). Stratified by domain, the mean (SD) scores were SP, 8.5 (±1.4); HH, 7.4 (±1.26); and HAIs, 5.7 (±1.55) (*p* < 0.001). The curriculum was reported as the most important source of knowledge by most students (86.7% for SP, 94.5% for HH, and 95.3% for HAIs), while self-learning was less frequently cited (11.7% for HH, 13.6% for SP, and 19.0% for HAIs). Additional sources included bedside training and hospital ward teaching (reported by 38.9% of students for HAIs, 65.2% for SP, and 68.8% for HH).

In the pilot study by Colosi et al. [48], which aimed to assess the level of knowledge of nursing and medical students regarding HAIs prevention and to identify differences between the two groups, nursing students achieved a higher mean score (18.5, SD ± 3.3) than medical students (17.4, SD ± 3.9), although the difference was not statistically significant (*p* = 0.186). In the SP domain, nursing students scored 6.92 (SD ± 1.37), in HH 5.28 (SD ± 1.2), and HAIs 4.45 (SD ± 3.15). HH knowledge differed significantly between nursing and medical students (*p* = 0.031), favoring nursing students. For both groups, significant differences were found between SP and HH (*p* < 0.0001) and between SP and HAIs (*p* < 0.0001).

The cross-sectional study by Alriyami et al. [34], assessed nursing students’ knowledge of HAIs and identified their main sources of information. Of 330 respondents, only 51 students (15.5%) achieved a knowledge score of 70 or above. The mean total score was 51.53 (SD ± 0.89) out of 100, with scores ranging from 16 to 96. Female students scored higher (53.84) vs. males (45.03), *p* < 0.05. The most frequently reported primary source of information was the Internet (46.4%), followed by clinical experience (39.8%). Mean scores varied significantly according to the main source: clinical experience (M = 3.53, SD = 14.76), curriculum (M = 5, SD = 12; *p* = 0.003), and Internet (M = 48.6, SD = 16.4; *p* = 0.022).

The cross-sectional study conducted by Brosio et al. [31] evaluated nursing students’ knowledge of HAI risk and prevention measures over the three years of their program, focusing on HH practices and SP adherence. Students from all years correctly identified age and invasive procedures as risk factors for HAIs (*p* = 0.0044 and *p* = 0.0117, respectively). Awareness of the environment as a potential source of infection increased with training progression (*p* < 0.0001). Third-year students were less likely to answer correctly about mortality associated with HAIs than first- and second-year students (*p* = 0.0130). No differences emerged regarding knowledge of HAI prevalence rates. SP adherence and awareness of contamination risks from biological fluids were consistently high across all years (approaching or reaching 100%). Appropriate glove use was well known, except for the recommendation to wear gloves for all procedures, which was correctly indicated by only half of third-year students (*p* < 0.0001). Knowledge of alcohol-based hand rub use was low: fewer than half knew it could replace antiseptic handwashing, and knowledge about its potential to substitute surgical handwashing was poor, with significant differences across years (*p* < 0.0001). Overall, HAI knowledge scores were 5.5/10 for first-year students, 6.4/10 for second-year students, and 6/10 for third-year students. SP scores were higher (8.8/10, 9.1/10, and 9.2/10, respectively), while HH scores were 6.7/10, 6.3/10, and 7/10. Authors set 7/10 as the arbitrarily acceptable score per domain.

The cross-sectional study by D’allessandro et al. [30] found that nursing students (mean 18.6, SD ± 2.9) scored significantly higher than medical students (mean 17.4, SD ± 3.5) on HAI prevention knowledge (*p* < 0.001). Stratification by domain confirmed significant differences (*p* < 0.001). Based on minimum acceptable scores defined in the protocol, the threshold (17.5 overall) was reached by all healthcare students combined, but when stratified by curriculum, only nursing students met the criterion. For HAIs (acceptable score = 3.5) and HH (acceptable score = 5.6), both groups scored below acceptable levels, while for SP (acceptable score = 8.4) both achieved appropriate scores.

In the cross-sectional study by Majidipour et al. [31] conducted to assess nursing students’ knowledge and performance regarding HAI control standards, the mean knowledge score was 12.49 (SD ± 2.3) out of 18 (95% CI, 11.83–12.89). Male students scored 13.00 (SD ± 2.6) and females 11.88 (SD ± 1.8), with no statistically significant difference. Third-year students scored slightly higher (12.82, SD ± 1.9) than fourth-year students (12.14, SD ± 2.7), but the difference was not significant. The mean performance score was 43.07 (SD ± 0.67) out of 54 (95% CI, 41.83–44.32), with female students performing significantly better than males (*p* = 0.014). Third-year students outperformed fourth-year students (44.6, SD ± 3.8 vs. 41.45, SD ± 7.3).

The quantitative observational pre–post study by Bouget Mohammedi et al. [33], evaluated nursing students’ knowledge of SP and HH before and after an IPC training program. Pre-test data were collected from 3739 students and post-test data from 2378. The mean baseline score (35.67/50) increased to 37.55/50 after training. Practical auditing was the only teaching method associated with a significant improvement (*p* = 0.050), while the presence of non-academic factors (NAD) increased the odds of achieving higher scores by 157-fold. The overall mean improvement was +2.1 points out of 50 (*p* < 0.001).

The cross-sectional study by Laiba Mazhar et al. [36] assessed BSN nursing students’ knowledge and practices regarding SP for infection control. Among the 64 participants, 91% recognized the importance of handwashing after each contact, 86% understood SP use with HCV-positive patients, but 63% erroneously believed they were necessary only for infected patients. Practices were good for handwashing (89%) and glove use (82%), but poor for protective eyewear use (72%) and vaccination (75%). Overall, 64% demonstrated adequate knowledge, albeit with critical gaps, and generally good but inconsistent practices.

The cross-sectional study conducted by Syed and Al-Rawi [37] assessed hand hygiene knowledge and practices among first-entry nursing students in Saudi Arabia (*n* = 304). The results showed a high awareness of healthcare-associated infection (HAI) transmission routes (94.2%), of the effectiveness of handwashing (93.2%), and of the necessity to perform hand hygiene before and after patient contact (83.2%). However, only 59.9% reported regular use of alcohol-based solutions. Significant gender differences emerged, with female students achieving higher mean scores both in knowledge (10.09 ± 1.27 vs. 9.63 ± 1.48; *p* = 0.004) and practice (5.00 ± 1.25 vs. 4.62 ± 1.46; *p* = 0.037).

The descriptive study by Shrestha et al. [38], conducted on 163 nursing students, explored knowledge, perceptions, and confidence regarding infection prevention and control (IPC) measures. Overall knowledge was rated as “fair” (71%), with higher scores for general IPC principles (85%) and lower scores for waste management (2%) and aseptic techniques (52%). Students’ perceptions were generally positive (mean 4.33/5) and strongly correlated with confidence (r = 0.781; *p* < 0.001). Knowledge varied significantly by year of study (*p* < 0.05), and significant correlations were found between knowledge and confidence (r = 0.343; *p* < 0.001) as well as between perception and knowledge (r = 0.329; *p* < 0.001).

Sharma and Bachani [39] compared knowledge, attitudes, practices, and perceived barriers regarding standard precautions between medical and nursing students in India (*n* = 200). Nursing students demonstrated better knowledge (57% vs. 48%), more positive attitudes (63% vs. 56%), and greater compliance with practices (42% vs. 38%), although the differences were not statistically significant (*p* > 0.05). The most frequently reported barriers included lack of personal protective equipment (PPE), heavy workload, and insufficient training.

In the observational study by Thakker and Jadhav [40], conducted on 198 students (84 medical, 74 dental, 40 nursing), hand hygiene knowledge was generally modest. Only 7.6% demonstrated good knowledge (>75% correct answers), while 69.2% achieved moderate scores and 23.2% low scores. Major gaps concerned the minimum recommended alcohol-rub duration and the correct choice of methods. Mean scores differed significantly between medical and dental students (*p* = 0.006) and between medical and nursing students (*p* < 0.001), but not between nursing and dental students (*p* = 0.068).

The study conducted by Darawad and Al-Hussami [49] examined knowledge, attitudes, and compliance with infection control precautions among 114 nursing students in their 3rd and 4th years. The results showed insufficient knowledge (mean = 49.64%, SD = 13.08). Despite clinical training, students reported low levels of adherence, highlighting the need for more effective educational strategies.

Wu et al. [50] in a cross-sectional study conducted on 175 fourth-year students of a five-year nursing program, assessed knowledge, application skills, and confidence regarding standard and additional precautions. While students showed good competencies in standard precautions (e.g., sharps disposal 98.3%, mask/goggle use 98.3%), severe gaps emerged in additional precautions (e.g., HIV 13.8%, mask with cough 13.2%). The overall mean score was 8.69/15 (SD = 1.55). Students with one month of training scored higher (*p* = 0.025), while paradoxically, those without short courses outperformed those who had attended them (*p* = 0.017).

Al-Rawajfah and Tubaishat [41] conducted a web-based survey among 594 nursing students from several Jordanian universities to explore knowledge and practices related to standard precautions. Results revealed an overall good level of knowledge (mean = 76.6% correct answers), with high scores in several domains but areas of uncertainty regarding tear secretions (49.8%), sweat (62.1%), and mask use in measles or chickenpox (58.8%). Classification indicated that 9.1% of students had weak knowledge, 39.6% satisfactory, and 51.3% excellent.

Ojulong et al. [42] investigated knowledge and attitudes regarding infection prevention and control (IPC) among health sciences students in Namibia, including 150 nursing students within a total sample of 385. Nursing students achieved a mean of 64% correct answers, with strong performance in glove use (85%) and sharps disposal (78%), but lower scores in hand hygiene (51%) and mask use (44%). Overall, nursing students scored significantly lower than medical students (*p* < 0.05).

The study by Rahiman et al. [51], conducted in South Africa on 301 nursing students from the second to the fourth year, assessed knowledge, attitudes, and practices regarding standard and transmission-based precautions. Findings showed a mean knowledge score of 8.12 ± 1.7/10, with 47.4% of students achieving high scores, 27.0% moderate, and 25.6% poor. Knowledge was stronger in standard precautions (60–90% correct answers) than in transmission-based precautions, where only 23% knew when to apply them and 86% were unaware of the concept of “protective environment.”

Blomgren et al. [43] compared hand hygiene knowledge among Swedish nursing students in their first and final semesters (*n* = 201). Using the WHO questionnaire, first-semester students scored a mean of 17.0 ± 2.1/25, while final-semester students scored 18.8 ± 1.8/25. Overall, knowledge levels were classified as moderate (55.7%), with a significant improvement observed as students progressed through their studies.

Khubrani et al. [44] assessed knowledge of standard precautions and infection control among 129 health sciences students, including 14 nursing students. Among this subgroup, 92.9% showed sufficient knowledge. Strongest areas included general concepts (85.7%), hand hygiene (72.8%), and PPE use (73.8%), while lower scores were recorded in sharps management (57.5%) and healthcare worker care (46.3%).

Bello et al. [45] compared levels of knowledge on nosocomial infections among health sciences students in Ghana (*n* = 200), including 70 nursing students. Nursing students demonstrated a moderate level of knowledge (61.3% ± 2.3), with higher scores in standard precautions (79.9%) and lower scores in hand hygiene (56.6%) and general knowledge of nosocomial infections (47.6%).

Mitchell et al. [46] explored knowledge of standard and transmission-based precautions among 349 final-year nursing students across six Australian universities. The overall correct response rate was 59.8% (95% CI 58.8–60.8), with a clear contrast between standard precautions (88.9% correct) and transmission-based precautions (27.2% correct; *p* < 0.001).

Finally, the national online survey by Gould and Drey [47], conducted with 488 preregistration nursing students who were members of the Royal College of Nursing, explored IPC experiences during clinical placements. All students reported witnessing episodes of non-compliance, most frequently related to lack of hand hygiene (76.4%), wearing jewelry (61.4%), nail polish or artificial nails (60%), breaches of isolation protocols (59.3%), PPE misuse (53.6%), and poor sharps management (52.3%). Despite reporting good theoretical preparation, many students highlighted confusion in applying certain procedures, particularly regarding isolation and aseptic sequence.

## 4. Discussion

This systematic review aims to investigate nursing students’ knowledge of healthcare-associated infections, which represent a significant issue in healthcare and nursing education.

Due to the nature of their training, nursing students are exposed daily to high risk environments and frequent contact with patients, family members, and healthcare workers, which makes them more vulnerable to infections [52].

They are estimated to be about 40% more susceptible to infections than the general population [53]. As pointed out by Tapias-Vargas et al. [54], a thorough understanding of exposure mechanisms, transmission risks, and prevention strategies is essential for students and healthcare professionals to actively contribute to the creation of a safe care environment.

Several studies have confirmed that nursing students possess a good basic knowledge of standard precautions and hand hygiene [35,48]. For instance, Rahiman et al. [51] in Malaysia and Shrestha et al. [38] in Nepal also reported moderate to good knowledge of standard precautions and hand hygiene among nursing students, with stronger results among those with clinical experience. Another relevant finding from the literature, as shown by Syed and Al-Rawi [37] in Saudi Arabia and by Wu et al. [50] in China who found high theoretical awareness but persistent inconsistencies in practice, particularly regarding the correct use of personal protective equipment and environmental disinfection. Such inconsistencies between knowledge and practice have also been observed in other contexts. For example, a study conducted in Pakistan by Laiba Mazhar et al. [34] revealed that although more than 90% of nursing students acknowledged the importance of handwashing and 86% recognized the need to apply standard precautions to HCV-positive patients, 63% mistakenly believed that these measures were required only for infected individuals. Adherence was high for practices such as handwashing after exposure to biological fluids (89%) and the use of gloves during blood collection (82%), but lower for the use of eye protection (72%). Moreover, only 75% of students reported being vaccinated against preventable diseases, and many highlighted insufficient resources for infection control. This discrepancy between theoretical knowledge and practical behavior was also emphasized by Bello et al. [45] and Ojulong et al. [42], who found that while most nursing students were aware of standard precautions, fewer consistently applied them in clinical settings. Blomgren et al. [43] highlighted that students often understand infection control principles conceptually but struggle to apply them under real clinical pressures, which reinforces the need for comprehensive educational strategies to translate knowledge into practice. These patterns mirror the broader literature, which consistently reports moderate-to-good baseline knowledge of standard precautions and hand hygiene among nursing students across diverse settings, with higher scores among those exposed to clinical placements. Prior reviews likewise note that theoretical awareness tends to be stronger than procedural detail, particularly for correct PPE use and environmental hygiene, indicating a common educational need that transcends country context [55,56].

These findings underscore the urgent need for targeted educational interventions, aimed not only at consolidating knowledge but also at correcting misconceptions, promoting consistent use of PPE, and ensuring adequate access to resources and vaccinations.

The level of preparedness among nursing students is also strongly linked to the practical experience acquired through internships and simulation activities, which transform theoretical learning into operational skills. This aligns with the concept of “embodied knowledge” [57], whereby learning becomes more effective when theory is integrated with practice. In this regard, clinical experience plays a crucial role in consolidating classroom knowledge [58]. Comparable findings were observed by Mitchell et al. [46] in Australia, who reported that students’ self-efficacy in infection prevention improved with clinical exposure, and by Khubrani et al. [44], who found that participation in specific IPC workshops significantly enhanced both knowledge and confidence levels.

Several studies have shown that while students tend to report confidence in their infection prevention practices, these findings are based on self-reported data rather than direct observation. For instance, both Tavolacci et al. [32] and Brosio et al. [35] assessed students’ knowledge and perceived compliance with infection prevention measures through questionnaires, without evaluating actual performance in clinical settings. Their understanding of more complex theoretical aspects, however, remains partial, particularly during the early years of training [25,59]. Taken together, this evidence highlights the need for an integrated educational approach that links theoretical instruction with supervised clinical experience and objective assessment of practice.

The academic curriculum is the main source of knowledge [32], but training on HAIs often lacks continuity, potentially compromising safety skills. Moreover, its integration into nursing programs remains uneven, indicating that the current academic model may not ensure full practical competence [51,60,61].

Several studies [30,32,48] show that nursing students score higher than medical students or students from other healthcare courses in areas related to hand hygiene, standard precautions and nosocomial infections.

Collectively, the findings of Colosi et al. [48], D’Alessandro et al. [30], and Tavolacci et al. [32] suggest that nursing students tend to achieve higher levels of knowledge in areas such as hand hygiene and standard precautions compared with medical students.

These results are consistent with those reported by Erasmus et al. [62] according to whom nurses, due to the nature of their role and their direct and continuous contact with patients, develop greater awareness and attention to infection prevention practices than other healthcare professionals. Similarly, Aloush (2017) and Carter et al. (2017) [60,63] highlighted that nursing training programs devote more hours to infection control than medical courses, which may explain some of the differences observed. However, Labrague et al. [22] emphasized that, despite higher knowledge levels, nursing students still engage in behaviors that deviate from guidelines, underscoring the persistent gap between theory and practice.

This gap is particularly evident in hand hygiene. Tavolacci et al. [32] reported only a partial understanding of its importance after glove removal, with 85.9% of students answering correctly. Similarly, Brosio et al. [35] found good general knowledge of standard precautions but limited awareness regarding proper glove use. These findings highlight the need to strengthen education on hand hygiene, which the World Health Organization [20] recognizes as the most effective measure for preventing the transmission of microorganisms. As noted by Bergamini et al. [64], a key issue lies in the perception of gloves as a form of self-protection rather than as a measure to prevent infection.

Gender differences emerging in various studies are a further element of interest. In general, female nursing students reported higher levels of knowledge than their male counterparts [31,34]. These elements are part of a broader picture of female academic success documented at international level [65,66], suggesting that the differences observed in knowledge of HAIs may reflect more general educational and attitudinal dynamics.

Nursing students’ level of preparedness appears to improve progressively as they advance through their training.

In Italy, D’Alessandro et al. [30] found higher average scores among third-year students than first-year students, indicating an improvement in knowledge with increasing academic experience. Brosio et al. [31] confirm this trend, observing a progressive increase in knowledge of HAIs in relation to the years of study, attributable to greater exposure to care settings and practical learning. Internationally, Alriyami et al. [34] report that, in a four-year course, fourth-year students achieve higher knowledge scores than second-year students, thanks to both advanced theoretical content and clinical experience. A similar trend was observed in Malaysia, where a strong and significant association emerged between academic year and level of knowledge, with more advanced students demonstrating more solid preparation in medical equipment disinfection and hospital infection prevention [67]. Further confirmation comes from a study conducted in Croatia, where third-year students demonstrated significantly higher knowledge than first- and second-year students regarding sepsis, one of the main complications of HAIs [68].

Student age is another factor associated with differences in knowledge. D’Alessandro et al. [30] found that students aged 24 years or older were significantly less likely to achieve acceptable levels of knowledge about standard precautions and hand hygiene than their younger colleagues. In Malaysia, the study by Abdul Wahab and Mohd Adie [69] reported a significant association between age and both knowledge and compliance with standard precautions, with better results among older students, likely due to their greater clinical experience. A survey conducted in Pakistan by Shukhali et al. [70] did not find significant differences in infection control knowledge scores between age groups, suggesting that the influence of age may vary according to context. The influence of age and training year on knowledge and compliance was further confirmed in studies by Rahiman et al. [51] and Syed and Al-Rawi [37], where older or more advanced students demonstrated greater adherence to standard precautions and better understanding of infection control principles.

In addition to demographic factors such as age, several studies have also explored the relationship between knowledge and performance in infection prevention and control. Majidipour et al. [31], in a cross-sectional study of third- and fourth-year nursing students, found a positive and significant correlation between these two aspects, with better results associated with a higher level of knowledge.

These data are in line with numerous studies highlighting that greater knowledge of HAIs is associated with stricter adherence to protocols, appropriate use of standard precautions and more solid practical skills in a clinical context.

A further finding concerns the impact of teaching methods and learning sources on nursing students’ preparedness for HAIs. Recent evidence reinforces the importance of experiential and interactive learning. For example, Rahiman et al. [51] and Blomgren et al. [43] demonstrated that scenario-based simulation and supervised clinical practice effectively bridge the gap between theory and practice. Similarly, Wu et al. [50] found that blended learning models combining e-learning with clinical mentorship improved both satisfaction and retention of IPC knowledge.

Alriyami et al. [34] observed that most students use the internet as their main source of information, followed by clinical experience, while the academic curriculum is mentioned less frequently. In contrast, Tavolacci et al. [32] found that the teaching program was the main source, followed by self-learning.

The literature highlights that training effectiveness does not depend solely on the source, but also on the teaching method and the degree of integration of infection control into the curriculum. Active and practice-oriented models, such as scenario-based simulation, are associated with higher levels of knowledge and practical skills acquisition, as well as greater student satisfaction. E-learning can also be effective, especially when combined with clear learning objectives and practical application opportunities [71,72,73,74,75].

Chang et al. [76] showed that structured and supervised teaching is more effective than self-directed learning. Da Silva et al. [26] emphasized the importance of grounding training in validated guidelines to ensure updated and accurate information. In France, Mohammedi [33] evaluated an HAIs course introduced in the first semester of the program. The course, based on multiple teaching methods, led to a significant increase in knowledge of standard precautions and hand hygiene. This demonstrates that early and contextualized training can produce immediate and measurable improvements. Conversely, the lack of curriculum integration and regular training may result in incomplete or outdated knowledge.

In conclusion, this systematic review has highlighted that, although nursing students demonstrate good knowledge on some key aspects of healthcare-associated infection prevention, theoretical gaps and differences related to individual and training factors remain. The overall analysis indicates that knowledge improves with advancing academic training, clinical experience and the adoption of active and contextualized teaching methods. However, the gap between theoretical knowledge and practical application persists, requiring more structured, integrated training interventions anchored to validated guidelines. In line with the objective of this review, the results provide useful guidance for the development of educational programs aimed at enhancing students’ infection control skills, with the ultimate goal of improving patient safety and quality of care.

### 4.1. Implications and Recommendations for Further Research

The findings of this review highlight the need to develop undergraduate nursing curricula that provide comprehensive and balanced preparation on HAIs, integrating theoretical content and practical activities in an equitable manner. As HAIs remain one of the major safety challenges in healthcare, student training should be firmly anchored to validated international guidelines, such as those issued by the World Health Organization (WHO) and the Centers for Disease Control and Prevention (CDC), to ensure that the knowledge conveyed is consistent, up to date, and applicable in real clinical contexts.

The evidence from this review clearly supports the need for nursing programs to adopt a structured and ongoing approach to teaching infection prevention and control (IPC). This should begin in the first year of study and be reinforced throughout subsequent semesters through a combination of theoretical lessons, practical simulations, and supervised clinical experiences. It is essential to clearly define core IPC competencies such as hand hygiene, standard precautions, and the appropriate use of personal protective equipment across all courses and to assess them through both formative and summative evaluations. To ensure consistency and sustainability, universities should align their curricula with established international guidelines, such as the WHO Core Components of IPC Programmes and CDC educational standards.

Particular attention should be given to active learning strategies, such as realistic scenario-based simulations on HAI prevention, hand hygiene workshops, and supervised training on standard precautions, to facilitate the transition from theoretical understanding to the safe and systematic application of prevention practices.

From a research perspective, there is a need for longitudinal and interventional studies that assess not only the immediate acquisition of IPC knowledge but also the ability to retain and apply such competencies over time, especially under conditions of clinical pressure. Future studies should adopt standardized and validated assessment tools that allow direct comparisons across contexts and support the development of robust and reliable meta-analyses.

It would also be important to explore how demographic and experiential factors, such as age, gender, number of internship hours, and type of clinical exposure, influence the acquisition and consolidation of HAI-related knowledge. Given the growing adoption of e-learning platforms, future research should examine the comparative effectiveness of online, blended, and traditional training methods, assessing their impact not only on knowledge but also on prevention practices in real-world settings.

### 4.2. Strengths and Limitation

This systematic review was conducted using a transparent and rigorous methodology, including a broad multi-database search, clearly defined selection criteria, and independent quality assessment. The use of standardized critical appraisal tools ensured that the synthesis was based on reliable evidence. Furthermore, the inclusion of studies from diverse geographical and socio-economic contexts provides a comprehensive view of nursing students’ knowledge of HAIs.

In line with PRISMA recommendations, the limitations have been explicitly organized to address potential sources of bias, study heterogeneity, and constraints in evidence synthesis.

Therefore, several limitations should be acknowledged. The included studies often involved small sample sizes, possible selection bias, lack of follow-up, and considerable methodological heterogeneity, all of which reduce the overall strength of the evidence. The review process was also limited by linguistic and temporal criteria, as well as by the exclusion of gray literature. The decision to exclude publications not written in English should be recognized as a potential source of bias. Additionally, the review protocol was not prospectively registered on PROSPERO, which may limit transparency and reproducibility. Moreover, due to the variability of study designs and outcomes, a narrative synthesis approach was adopted, and the overall quality of evidence suggests that the conclusions should be interpreted with caution. It should also be noted that all studies were included regardless of their quality scores, as none showed a risk of bias significant enough to compromise their overall validity or relevance to the research question. This decision allowed the inclusion of studies with lower scores, which nonetheless contributed valuable and complementary insights to the synthesis, enriching the understanding of the topic across different methodological and geographical contexts.

The heterogeneity of study designs, assessment tools, and knowledge scoring systems limited the possibility of direct comparison and meta-analysis. In addition, most of the included studies were cross-sectional, which prevents establishing causal relationships between training, knowledge, and practice. Variability in curriculum structures and infection control education across countries may also limit the generalizability of the findings. Self-reported data in several studies may be subject to recall bias or social desirability bias, potentially leading to an overestimation of knowledge levels or adherence to measures for the prevention of healthcare-associated infections. Overall, these elements indicate that the findings of the review should be interpreted with caution, but at the same time provide a useful and coherent basis to guide future research and educational interventions.

## 5. Conclusions

This systematic review highlights that nursing students’ knowledge of HAIs prevention is often fragmented and insufficiently consolidated, with relevant theoretical gaps and inconsistencies in practical application. Evidence from the included studies shows that knowledge tends to improve with academic progression, clinical experience, and the adoption of active, contextualized teaching methods anchored to validated international guidelines. To strengthen preparedness, nursing curricula should integrate theory and practice more systematically, ensuring that infection prevention competencies are comprehensive, up to date, and consistently applied in clinical contexts. Enhancing HAI-related education is crucial to equip future nurses with the skills needed to safeguard patient safety and improve the quality of care.

## Figures and Tables

**Figure 1 ijerph-22-01609-f001:**
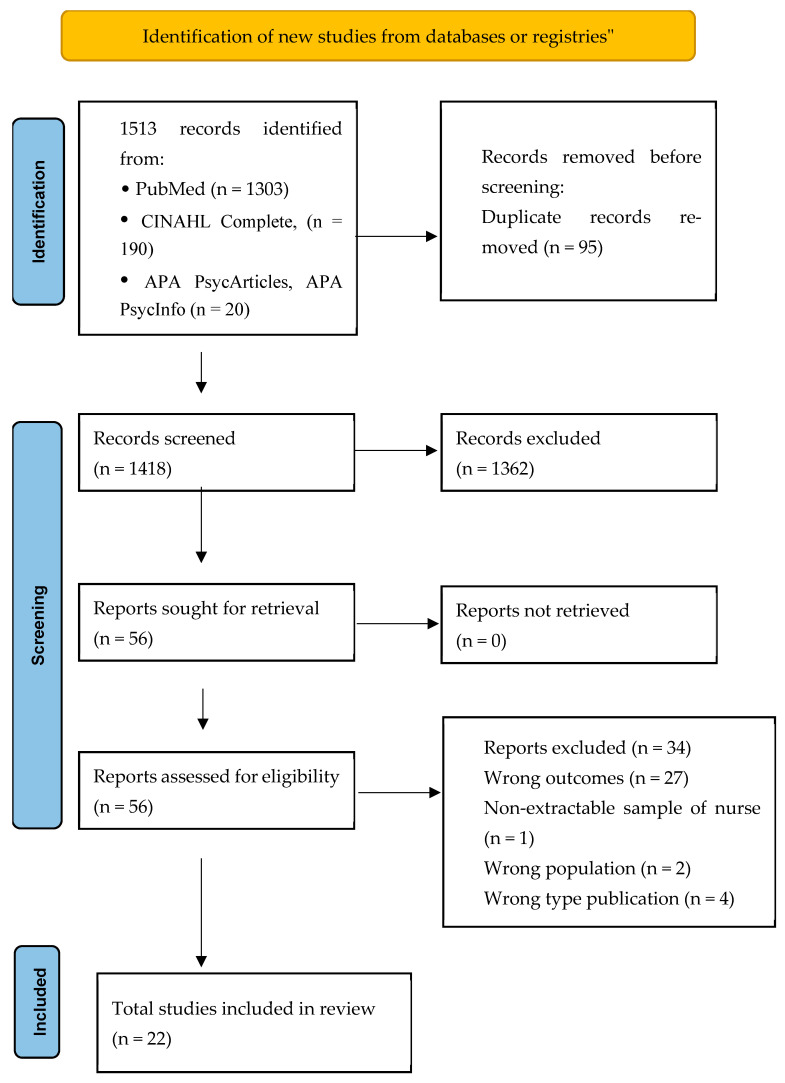
PRISMA 2020 flow diagram model for systematic reviews’ [27].

**Table 1 ijerph-22-01609-t001:** Research question based on the PEO framework.

P	Nursing students
E	Healthcare-associated infections (HAIs)
O	Knowledge of HAIs

**Table 2 ijerph-22-01609-t002:** PubMed Query.

Database	Search String
PubMed	(“students, nursing” [MeSH Terms] OR (“students” [All Fields] AND “nursing” [All Fields]) OR “nursing students” [All Fields] OR (“nursing” [All Fields] AND “student” [All Fields]) OR “nursing student” [All Fields] OR ((“undergraduate” [All Fields] OR “undergraduate s” [All Fields] OR “undergraduated” [All Fields] OR “undergraduates” [All Fields]) AND (“student s” [All Fields] OR “students” [MeSH Terms] OR “students” [All Fields] OR “student” [All Fields] OR “students s” [All Fields]) AND “nurs *” [All Fields])) AND (((“cross” [All Fields] OR “crosse” [All Fields] OR “crossed” [All Fields] OR “crosses” [All Fields] OR “crossing” [All Fields] OR “crossings” [All Fields]) AND “infect *” [All Fields]) OR (“cross infection” [MeSH Terms] OR (“cross” [All Fields] AND “infection” [All Fields]) OR “cross infection” [All Fields] OR (“healthcare” [All Fields] AND “associated” [All Fields] AND “infection” [All Fields]) OR “healthcare associated infection” [All Fields]) OR ((“hospital s” [All Fields] OR “hospitalisation” [All Fields] OR “hospitalization” [MeSH Terms] OR “hospitalization” [All Fields] OR “hospitalised” [All Fields] OR “hospitalising” [All Fields] OR “hospitality” [All Fields] OR “hospitalisations” [All Fields] OR “hospitalizations” [All Fields] OR “hospitalize” [All Fields] OR “hospitalized” [All Fields] OR “hospitalizing” [All Fields] OR “hospitals” [MeSH Terms] OR “hospitals” [All Fields] OR “hospital” [All Fields]) AND “infect *” [All Fields]) OR ((“nosocomial” [All Fields] OR “nosocomially” [All Fields] OR “nosocomials” [All Fields]) AND “infect *” [All Fields])) AND (“knowledge” [MeSH Terms] OR “knowledge” [All Fields] OR “knowledge s” [All Fields] OR “knowledgeability” [All Fields] OR “knowledgeable” [All Fields] OR “knowledgeably” [All Fields] OR “knowledges” [All Fields] OR (“awareness” [MeSH Terms] OR “awareness” [All Fields] OR “aware” [All Fields] OR “awarenesses” [All Fields]) OR (“compete” [All Fields] OR “competed” [All Fields] OR “competences” [All Fields] OR “competencies” [All Fields] OR “competently” [All Fields] OR “competents” [All Fields] OR “competes” [All Fields] OR “competing” [All Fields] OR “mental competency” [MeSH Terms] OR (“mental” [All Fields] AND “competency” [All Fields]) OR “mental competency” [All Fields] OR “competence” [All Fields] OR “competency” [All Fields] OR “competent” [All Fields]) OR “attitude *” [All Fields] OR “perception *” [All Fields] OR “practice *” [All Fields])

The asterisk (*) indicates a truncation symbol used in database searches to include all possible word endings.

**Table 3 ijerph-22-01609-t003:** CINAHL Query.

Database	Search String
CINAHL	(Nursing students or student nurses or undergraduate student nurses or pre-licensure nurse) AND (cross infection or hospital acquired infections or healthcare associated infections) AND (knowledge or education or understanding or awareness or competence OR attitude or perception * or practice *)

The asterisk (*) indicates a truncation symbol used in database searches to include all possible word endings.

**Table 4 ijerph-22-01609-t004:** APA PsycArticles and APA PsycInfo Query.

Database	Search String
APA PsycArticles APA PsycInfo	(Nursing students or student nurses or undergraduate student nurses or pre-licensure nurse) AND (cross infection or hospital acquired infections or healthcare associated infections) AND (knowledge or education or understanding or awareness or competence OR attitude or perception * or practice *)

The asterisk (*) indicates a truncation symbol used in database searches to include all possible word endings.

## Data Availability

Data are contained within the article.

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
