# Peer review of "Nursing Students’ Knowledge Among Healthcare-Associated Infections: A Systematic Review"

_ijerph, 2025, doi:10.3390/ijerph22111609_

Round 1
Reviewer 1 Report
Comments and Suggestions for Authors
Dear authors,
Thank you for the opportunity to review your manuscript. The paper addresses a relevant issue, given the persistent burden of healthcare-associated infections and the central role of nursing students in infection prevention and control. The review is well-structured, follows PRISMA guidance, and includes a transparent synthesis supported by validated appraisal tools.
At the same time, I would encourage you to strengthen several aspects:
1. Search strategy – While you searched multiple databases, the search strings appear too restrictive, especially regarding the outcome terms. In PubMed, for example, only knowledge was used, excluding important synonyms such as awareness, competence, attitudes, perceptions, practices. This may have led to the omission of relevant studies. Expanding the outcome terms, combining MeSH and free-text keywords, and applying truncation (e.g., educat*, infect*) could improve comprehensiveness. The decision to exclude non-English publications should also be acknowledged as a source of potential bias.
2. PRISMA reporting – Although the manuscript generally follows PRISMA 2020, several items are incomplete or missing:
-
No evidence of protocol registration (e.g., PROSPERO or OSF), which would increase transparency.
-
Lack of search dates and full reproducible search strategies for all databases.
-
No formal assessment of reporting bias (e.g., publication bias).
-
The limitations of evidence section could be more explicitly structured in line with PRISMA items.
3. Quality Appraisal - The use of JBI and NIH tools is a strong methodological choice. However, it is not fully clear how the results of these assessments were incorporated into the synthesis (e.g., whether studies rated as moderate quality had a different weight in the conclusions). Clarifying this integration would strengthen the credibility of the findings.
4. Implications for curricula – The discussion highlights curricular variability but would benefit from more concrete recommendations on how to systematically integrate infection prevention and control training into nursing education.
Overall, this manuscript provides valuable insights and makes an important contribution. By refining the search strategy, aligning more closely with PRISMA reporting standards, and streamlining the narrative, you will further enhance its methodological rigor and clarity.
Reviewer 2 Report
Comments and Suggestions for Authors
Thank you for the opportunity to review this article.
This article presents a systematic review exploring nursing students' knowledge and awareness of healthcare-associated infections (HAI). It identifies existing gaps in understanding with the aim of informing and enhancing nursing curricula.
1. Introduction - this section is done very well with one recommendation.
The authors have provided a clearly constructed introduction which logically flows from the problem of HAI and the importance of the issue. It then discusses the issue broadly from a global level, narrowing to region (Europe) and then country (Italy). It then sets the scene for the review by explaining the nursing focus and nursing education, discusses global recommendation for integration HAI knowledge into curriculum then clearly states the current status of HAI in Nursing education.
Line 84 - a gap in the text between two commas.
Line 93 - Sentence starting with 'In literature, there are....' requires restructuring for clarity.
2. Materials and Methods - this section is clearly written as to what the authors did, however the search strings and search outcomes are confusing.
The authors search outcome cites the number of articles identified through PubMed, then collectively cites the articles identified in CINAHL, MEDLINE (which does not appear in the PRISMA table), APA PsycArticles and APA PsycInfo databases collectively. This description also does not match the Tables associated with this section, as they only describe PubMed, CINAHL and APA Psych journals. This makes it difficult to duplicate the search and identify how many articles were captured from each database. when attempting to search using the cited search strings, the number of papers identified were vastly different.
Tables/Appraisal tools in this section are appropriate, and supported with a clear explanation in text.
Results - The text in this section mostly reads well, however there are some questions about the data extraction table (Table 7).
Line 64 - add the word country after the word Asian .
Line 302 - remove the text 'at the University if Ferrara' as it does not add anything to the text and the information can be found in Table 7
Line 337 - remove the words 'in France' for the same reasons as above
Line 344 - remove the words 'in Pakistan' for the same reasons as above.
Table 7
Colosi et al, (2011) - the Effect Estimate cites p=0.013, yet in the text associated with the article it cites a different p value of p=0.031. When reading the original text however, the abstract cites p=0.13, yet the text cites P=0.031 which is also different. Please either ascertain the proper p-value, or exclude the article due to inconsistency.
Brosio et al (2017) - It was difficult to interpret the Effect Estimate alongside the Main Results as the two columns didn't align well. For example 'Awareness of environment' as an infection source is not mentioned in the main results, yet has an Effect Estimate cited. Also it was difficult to ascertain from the Effect Estimate column which direction the effect was as only a p-value is cited without any other information.
D'Alessandro et al. (2014) - Main results column is comparing nursing students to medical students - when citing results by area (SP, HH and HAI) the p-value for significant differences between the two cohorts was cited however it does not say in which cohort's favour.
Mohammedi et al (2025) - the citation is inconsistent with the other journal articles in the table. The column titled identified limitations cites 'Conducted only in France' as a limitation. I would disagree with this as the sample size for this study was large and the aim of the study was to study nursing students in France only, so it was specific for France with a large sample size.
4. Discussion - This section requires a review. Many parts of the discussion read similarly to results/findings rather than discussion (lines 407 onwards). I would like to see this re-written to be integrate better with the broader literature. Themes are difficult to discern in some parts even though they are defined. This section lacked the fluidity of the other parts f the text which made reading and following the discussion more difficult.
Line 365 - What do the authors mean by 'main problems'? Given the review only looked at HAI knowledge in nursing students, there is no comparitor and there is also no in-text citation to draw information from as to ascertain what the other problems in health care are.
Line 395 - the authors have written "...while students demonstrate satisfactory practical skills..." - when in reality the articles that the authors have cited that supports this claim did not assess practical skills, only assessed knowledge with participants reporting skills - but were not actively observed, so this is a bold statement.
Paragraph beginning line 448 - It is unclear the relationship between the two lines of discussion within this paragraph. The line of discussion ending on line 457 with the word 'context', and the subsequent sentence beginning with 'Within'.. whilst the first part is discussing age of participants as a factor, the second part doesn't refer to age.
Line 472 - the sentence starting with 'Active...' appears to provide support from the broader literature around teaching methods and source of information from the previous section of the paragraph. However it brings in the CDIO framework as an example which seems to have little relevance to the rest of the paragraph.
Subsections 4.1 and 4.2 are written well.
5. Conclusion - clear and concise.
References - There were many references which would be considered not recent, however I would accept this given the aim and scope of the review. There are inconsistent referencing formats in lines 659 (capitalisation of article title) and the same for line 722.
Overall the article describes what it intended to. It supports the notion that nursing students' knowledge of HAI is inconsistent and adds to the evidence that consistency in curricula is required to help address this.
Round 2
Reviewer 1 Report
Comments and Suggestions for Authors
Thank you for your thorough revision. You have adequately addressed the previous comments, and the manuscript is now clearer.
One suggestion, consider moving some of the larger tables to the supplementary material, as this would improve the readability and flow of the main text.
Overall, this version represents a meaningful improvement — well done.
